# Sewage Sludge as Inhibitor of the Formation of Persistent Organic Pollutants during Incineration

**Juan A. Conesa** [1,2]

1    Department of Chemical Engineering, University of Alicante, P.O. Box 99, E-03080 Alicante, Spain; ja.conesa@ua.es; Tel.: +34-965-903-400
2    Institute of Chemical Process Engineering, University of Alicante, P.O. Box 99, E-03080 Alicante, Spain

**Abstract:** With the objective of suppressing dioxins and furans (PCDD/Fs) emission in municipal solid waste incineration plants (MSWI), different chemical inhibitors have been tested. Among these inhibitors, nitrogen and sulphur compounds can significantly suppress PCDD/Fs formation via de novo synthesis, which gives very interesting results with very little capital investment. In recent years, the possibility of using waste rich in nitrogen and/or sulphur as a source of inhibitor compounds has been considered, and thus has reduced the emissions of pollutants while the waste is treated. The effect of adding sludge from urban sewage treatment plants in three variants has been specially studied: directly mixing the waste, using the decomposition gas of the previously dried sludge, and using the decomposition gas of the sludge together with other inhibitors such as thiourea. Reduction of emissions in laboratory tests using model samples indicated the efficiency to be higher than 99%, using sewage sludge (SS) as an inhibitor whereas, in actual MSWI plants, the efficiency can be as high as 90%.

**Keywords:** PCDD/Fs; inhibition; POPs; combustion; wastes; co-combustion; co-incineration; dioxins

## 1. Introduction

It is a reality that the production of solid urban waste, although it has been decreasing in recent years, continues to be a problem, with a production in Europe in 2019 of 502 kg/person/year [1]. The culture of recycling of many kinds of wastes is growing more and more in society; however, landfills are still receiving greater loads of municipal wastes, even though community policy advocates a higher recycling rate and energy use of the rest fraction (not recyclable) in incineration furnaces.

In many industrialized countries, resources contained in waste are recovered by energy recovery as the main option. In this sense, it is necessary to make an effort to reuse or compost municipal solid waste, implementing a final step of energy recovery from the non-reusable and non-compostable matter.

The processes of thermal treatment of waste present a series of advantages over other methods, such as the reduction of waste (70% by mass and 90% by volume, on average), the inerting of waste (destruction of biological contamination and toxic organic compounds), the recovery of the calorific value of the waste, and the substitution of fossil fuels for the generation of Energy. However, the main problem of waste incineration is the possible production of pollutants that can cause damage to the environment.

Among the possible emitted pollutants, polycyclic aromatic hydrocarbons (PAHs) are a group of organic compounds, lipophilic in nature, solid at room temperature, and formed because of incomplete combustion. PAHs can be found in fuel-derived products such as oil, coal, and tar deposits and also as products of the use of fossil or biomass fuels. PAHs are of concern since many of them cause a variety of conditions, including cancer. PAHs have been classified into different groups, relating their toxic properties [2]. Martinez et al. [3] and Richter et al. [4], among others, conducted a review on the emission and generation of PAHs in energy-generating combustion processes and in the formation

of soot. Sánchez et al. [5] presented an analytical method to determine combustion PAHs both in the gas phase and in the particles. These pollutants are present also in recycled plastic items, and have been identified as causing possible negative effects on this type of materials [6].

Other dangerous compounds are emitted during incineration, as those known as persistent organic pollutants (POPs). POPs are different groups of compounds that remain intact in nature for years once they have been released into the environment, due to a combination of physical and chemical properties; also, by their nature, POPs accumulate in the fatty tissue of living organisms.

Usually, POPs are divided into two groups (see Figure 1): "legacy" POPs, i.e., substances with the particular properties of POPs that are long-recognized as harmful and controlled under international regulation since at least 2003, and "emerging" or "new" POPs that are being considered for banning purposes nowadays [7].

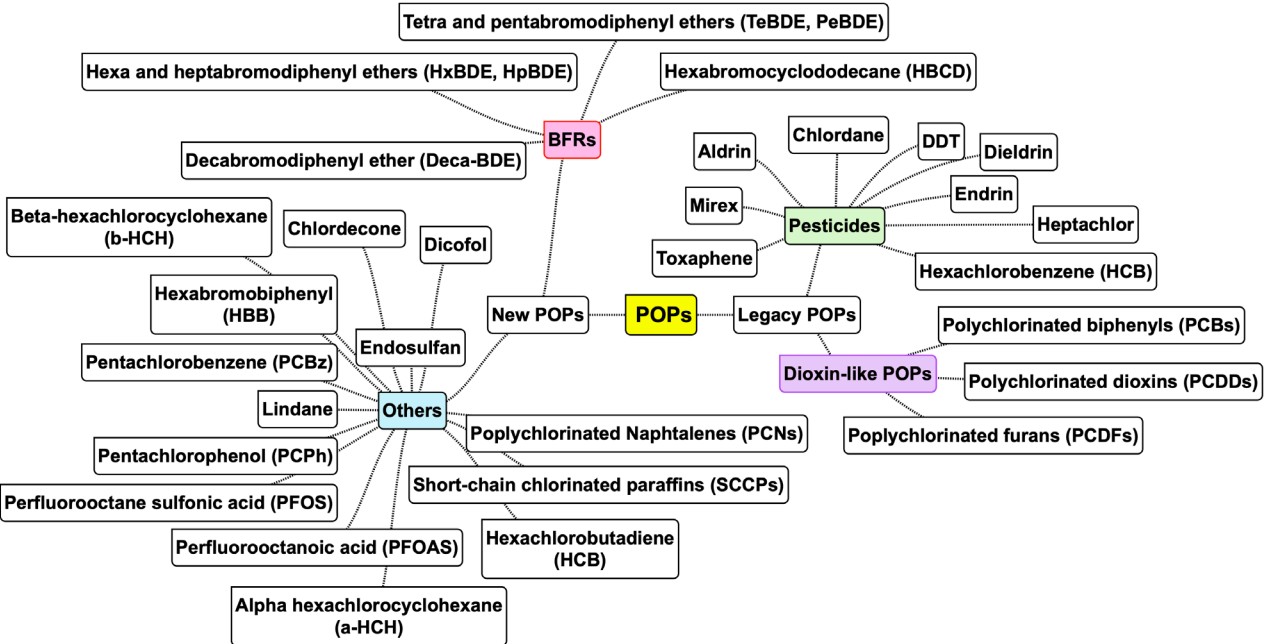

**Figure 1.** Persistent Organic Pollutants (POPs) considered in the Stockholm Convention [7].

In "legacy" POPs, pesticides and dioxin-like compounds are distinguished. The latter group, of particular importance given their toxicity, includes dioxins (PCDD), furans (PCDF), and biphenyls (PCB). On the other hand, the compounds classified as "new" POPs are also divided into two groups, consisting of a group of brominated compounds used as flame retardants (BFR), and another more heterogeneous group with substances that are pollutants by themselves or that can lead to very dangerous pollutants.

Among the POPs, the term 'dioxins' is applied to a set of aromatic substances, and usually refers both to PCDDs and PCDFs [8]. There are 75 chemical congeners for PCDDs and 135 congeners for PCDFs. Of all these compounds, there are 17 that develop toxic effects, and they are those that have chlorine atoms in positions 2, 3, 7, and 8. The toxicity of the 17 compounds is measured with an equivalent toxicity factor with respect to the most toxic (2,3,7,8-TetraCloroDibenzoDioxin).

PCBs (Poly Chlorinated Biphenyls) are a family of 209 compounds, of which about 130 have been identified in commercial mixtures that were industrially manufactured under different names (Aroclor, Clophen, Pyralene, etc.). Its commercialization and use have been restricted in Europe since the 85/467/EC regulation came into force. Certain PCBs, the so-called coplanar or mono-ortho congeners, are considered compounds closely related to dioxins and furans as they present similar toxicities.

There are several theories about the molecular mechanisms by which PCDDs and PCDFs are formed and subsequently emitted from combustion sources [8]. These theories are based on observations made in municipal solid waste incinerators and laboratory studies. From the studies and reviews carried out, various pathways could be considered, although the relative importance may be very different. Schematically (see Figure 2), it can be said that it can occur that the dioxins that come out through the flue gas, the fly ash, and the bottom ash come from the initial raw material, or are formed, either in the catalytic homogeneous or heterogeneous gas phase. In the homogeneous phase, they can be formed from organic compounds smaller than dioxins/furans, called precursors, that will be chlorinated derivatives, or they can be chlorinated by the radical Cl·; in the heterogeneous catalytic phase (chlorides of Cu, Zn, Fe, and others) chlorination is carried out mainly by means of $Cl_2$, from precursors or from carbonaceous structures (de novo synthesis). Oxygen can be incorporated into the precursors, while, in de novo synthesis, the carbonaceous structure must be oxidized [9–11].

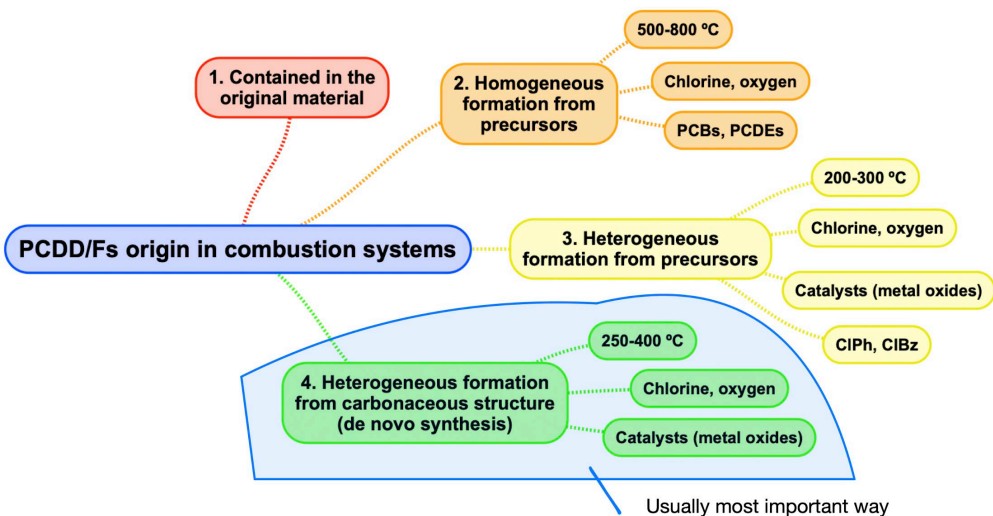

**Figure 2.** Possible pathways for PCDD/Fs formation in combustion systems, indicating the necessary conditions to be produced. De novo synthesis is usually the most important method, as it is the one presenting faster kinetics in the usual conditions of incineration systems [12].

Different studies [12,13] have highlighted the importance of the nature of waste (particularly, its content of Cu and Fe metals and chlorides) in the formation of dangerous compounds in the treatment processes. Likewise, the key role of the presence of oxygen in the production of PAHs, dioxins, and related compounds is highlighted, observing maximum amounts of these pollutants at intermediate oxygen levels.

Another aspect that may be interesting to analyze about the formation of dioxins in incinerators is the distribution of congeners, or what is also known as the fingerprint. By the de novo synthesis process, a distribution is obtained where the ratio between furans and dioxins is greater than unity. Fiedler et al. [10] observed that the distribution of congeners emitted in municipal waste incinerators and hazardous waste incinerators are indistinguishable. The most representative toxic congener in both cases is 23478-PeCDF, which contributes more than 30% to the total equivalent toxicity (I-TEQ).

In the present work, a review of data found in the literature about inhibition with chemicals in MSWI plants is done, with particular interest in the use of sewage sludge as a material that provides sulfur and nitrogen, and investigating the mechanism of action of these compounds.

## 2. Inhibition in the Dioxins and Furans Formation by S- and N-Containing Compounds

In various research, tests have been carried out showing the ability of various nitrogen and sulfur compounds to prevent the formation of dioxins and furans (PCDD/Fs) in the thermal destruction of waste. Already in 1998, Ruokojarvi et al. [14] studied the effect of four gaseous inhibitors (sulfur dioxide, ammonia, dimethylamine, and methyl mercaptan) on PCDD/PCDF formation in the combustion of liquid fuel. In a pilot plant scale, the authors observed a high decrease (98%) in the concentrations of PCDD/Fs when inhibitors were added, especially in the particle-phase PCDD/Fs concentrations.

In 2005, Pandelova et al. [15] studied the effect of the addition of more than 20 compounds to the co-combustion of coal and waste, finding that the most effective inhibitors for PCDD/F formation in flue gases were $(NH_4)_2SO_4$ and $(NH_4)_2S_2O_3$, both containing N and S. Table 1 shows a summary of the most important data noted in this Review.

Samaras et al. [16,17] mixed various sulfur and nitrogen compounds with RDF (refuse-derived-fuel) in proportions of 10% by weight. They found that the elimination of PCDD/Fs is higher than 98% using sulfur compounds, which combine with metals to form complexes and avoid the formation of pollutants. On the other hand, removal was somewhat lower using urea [16]; however, the way urea is added to the reaction medium does not affect its ability to prevent PCDD/Fs formation [17].

Thiourea $[(NH_2)_2CS]$, on the other hand, with a molar ratio of $(S + N)/Cl$ only 0.47, inhibits up to 97.3% of the PCDD/Fs content (99.8% in equivalent toxicity), although the study has not been carried out on an industrial scale as in the previous case [18]. As can be seen when comparing the results of ammonium sulfate with thiourea, the amino functional group $NH_2^-$ tends to be more effective than $NH_4^+$. This fact has been confirmed by several authors, and it is attributed to the fact that the $N{\equiv}C$ group appears in the decomposition of thiourea at 250–500 °C (in addition to ammonia, carbodiimide and $H_2S$), which can poison the metallic catalyst by forming some thermally stable compounds (metal complexes) [19]. The same is true of some sulfur compounds, which form sulfides or sulfates with the catalyst.

Skodras et al. [20] studied the combustion and co-combustion of natural wood, medium density fiberboard (MDF with nitrogenous additives), light poles (with a high metal content), and lignite, observing that the lower PCDD/Fs emissions are achieved using mixtures with MDF with nitrogenous additives.

Wu et al. [21] used $(NH_4)_2SO_4$ and pyrite $(FeS_2)$ to suppress dioxin formation in a commercial incinerator, showing that removal is only partial on a practical level. However, Amand et al. [22] showed that the addition of ammonium sulfate effectively reduces the emission in a fluidized bed incinerator. For their part, Fu et al. [18] used ammonium thiosulfate, sulfamic acid $(H_3NSO_3)$, and thiourea $(CH_4N_2S)$ to reduce the emission of PCDD/Fs, highlighting that the combination of sulfur and nitrogen compounds provide better inhibition than separately. Thiourea has been considered by various authors, by combining both atoms. Lin et al. [23] used this in a municipal incinerator with very low doses (0.1%), showing a 91% reduction in the emission. Furthermore, these authors indicated that a reduction of NOx-type compounds takes place simultaneously. Both factors are explained by the poisoning of the catalyst metals in the presence of thiourea.

Comparing the inhibitory effect of 21 groups of substances, Pandelova et al. [15] observed that adding compounds containing only sulfur or only nitrogen achieved a relatively low inhibitory effect, while, when adding sulfur and nitrogen compounds together, the effect was much more important. For example, by adding 1% by weight of sulfamic or sulfamidic acid $(H_2NSO_3H)$, a PCDD/Fs inhibition efficiency of around 96% was achieved [16]. Therefore, most recent studies focus on the use of compounds containing nitrogen and sulfur, where, in addition to sulfamic acid, thiourea, sulfate, and ammonium thiosulfate, among others, have been tested directly in commercial incinerators and with good results [18,21,23].

**Table 1.** Summary of data found in the literature.

| Reference | Main Fuel | Inhibitor Used | Main Finding (PCDD/Fs Inhibition Efficiency, %) |
|---|---|---|---|
| Amand et al. [22] | SRF and wood | SS | Important decrease in the gases emission, as well as in the filter and cyclone ashes |
| | | Ammonium sulphate | |
| Chen et al. [24,25] | Model samples | SS drying gases | >99%. for some congeners, especially for dioxins versus furans |
| Chen et al. [25] | Model fly ash | SS | 97.6% (similar to TUA) |
| Chen et al. [25] | Model fly ash | SS | 99% of inhibition in terms of toxicity (higher for dioxins than for furans) |
| | | $SO_2$ and $NH_3$ | $SO_2$ was more effective than $NH_3$ (61.9% and 38.6%, respectively) |
| Chen et al. [24] | Model fly ash | SS drying gases | 97.6%, $NH_3$ and $SO_2$ are the most important components of the SS drying gases |
| Chen et al. [26] | Fly ash from MSW | SS | >96% |
| Fu et al. [18] | Model fly ash | sulfur–amine/ ammonium compounds | The combination of sulfur and nitrogen compounds provide better inhibition than separately |
| | | | The amino functional group $NH_2^-$ tends to be more effective than $NH_4^+$. |
| | | | TUA (99:8%) > ASA (92:4%) > ATS (85:4%) |
| Gandon-Ros et al. [27] | PVC | SS, low $O_2$ presence | Increased PCDD/Fs emissions, non-inhibition observed due to the high percentage of metals in the SS |
| | | SS, higher $O_2$ presence | 89.2%, 71.4% and 98.8% for the inhibition ratios 0.25, 0.50 and 0.75, respectively |
| Hajizadeh et al. [28] | Model fly ash | $SO_2$ and $NH_3$ | $NH_3$ was more effective at the lower temperature, while the behavior of $SO_2$ was the opposite (reducing a higher proportion of PCDDs than $NH_3$ at the higher temperature) |
| Lin et al. [23] | MSW | 0.1% thiourea | 91% |
| Lin et al. [29] | SS | inorganic flocculants (poly-ferric chloride, poly-aluminum chloride) | Increased PCDD/Fs emission, 30% increase |
| | SS | organic flocculants (polyacrylamide) | 30–40% decrease |
| Lundin et al. [30] | Biomass and MSW | ammonium sulfate | 80% (PCDDs) 50% (PCDFs) 45% (PCBs) |
| Ma et al. [31] | MSW | CaO and S (sulphur) | S (sulphur): HpCDD/Fs inhibition 88.1% CaO: HxCDD/Fs inhibition 85.1% |
| Mi Yan et al. [32] | Fly ash | ammonium sulfate and urea | both inhibitors had a higher inhibition efficiency at high temperature (650 °C) than at low temperature (350 °C) |
| | | | greater reduction in furan emissions (PDCFs) |
| Moreno et al. [33] | PVC containing waste | PUF | PCDD/Fs 85.7% dl-PCBs 81.2% |
| Ogawa et al. [34] | Wood with PVC | Coal with S (sulphur) | significant reduction in PCDD/Fs, being somewhat more important for PCDFs |
| Pandelova et al. [15] | | 1% by weight of sulfamic or sulfamidic acid | 96% |

**Table 1.** *Cont.*

| Reference | Main Fuel | Inhibitor Used | Main Finding (PCDD/Fs Inhibition Efficiency, %) |
|---|---|---|---|
| Ruokojarvi et al. [14] | Liquid fuel | sulfur dioxide, ammonia, dimethylamine, and methyl mercaptan | 98% |
| Samaras et al. [16,17] | RDF | Sulphur compounds | >98% |
| | | Urea | Slightly lower than 98% |
| Soler et al. [35] | Model fly ash | TUA, TSA, ASA | the presence of the inhibitors accelerated the decomposition of the model fly ash Reactivity: TSA $\approx$ TUA > ASA |
| Wang et al. [36] | De novo runs | Different nitrogen containing compounds | $NH_4H_2PO_4$ > $NH_4HF_2$ > $(NH_4)_2SO_4$ > $NH_4Br$ |
| Xiao et al. [37] | Sawdust | SS | 62.9% for pellets with 10% SS 35.4% for pellets with 30% SS |
| Xiao et al. [37] | Wood sawdust | SS | there is a predominance of furans (PCDFs) over dioxins (PCDDs) |
| Zhang et al. [38] | Coal | SS | PCDD/F emissions increased from 7.00 to 32.72 pg I-TEQ/$Nm^3$ as the amount of SS increased from 5 to 20% |
| Zhan et al. [39] | Model fly ash | SS, high temperature (not de-novo synthesis) | >90% |
| Zhan et al. [40] | MSW | SS and TUA | >90% both |
| Zhong et al. [41] | MSW | 5% SS | 32% |

Among these inhibition studies using nitrogen and sulfur compounds, research done at the University of Umeå (Sweden) used ammonium sulfate $[(NH_4)_2SO_4]$ in the co-combustion of biomass and waste from the paper industry, which had a high chlorine content [30]. This compound was injected into the waste incineration plant after combustion and just before the cyclones, when the temperature was 800 °C, then the outlet gas was sampled at 130 °C. As a result, a reduction of over 80% was obtained in PCDDs, over 50% in PCDFs, and 45% in PCBs (although the latter were in a much lower proportion than the former). The inhibition was similar for all congeners.

Lin et al. [29] detected an increase in the PCDD/Fs production when adding inorganic flocculants to the incineration of SS, but an important decrease in the case of adding organic flocculants with amino substituents, ascribing the effect to the presence of such organic groups.

However, ammonia acts in a different way, since it reduces the concentration of hydrogen chloride, necessary for the formation of PCDD/Fs [42]; that is, it blocks chlorination through the formation of ammonium chloride. Larger nitrogen and sulfur molecules appear to condense and adsorb more easily in the reaction system to deactivate the catalyst than $NH_3$ and $SO_2$ [40]. In any case, as Fujimori et al. [43] found, the three inhibition pathways for dioxin formation are: (i) catalyst poisoning by sulphuration of the catalyst to form compounds of the type CuS, $CuS_2$, and $CuSO_4$; (ii) chlorination blocking, i.e., reaction of chlorine with nitrogen compounds that would prevent the chlorination of carbon; (iii) changing the surface of the carbonaceous matrix by sulfur and nitrogen attack to inhibit the formation of chlorinated aromatic compounds during de novo synthesis. Figure 3 shows the schematic of these three ways. Temperature plays a fundamental role in inhibition since, when using a higher temperature, thiourea tends to decompose into nitrogen and sulfur compounds instead of reacting with the metal catalyst [19,43].

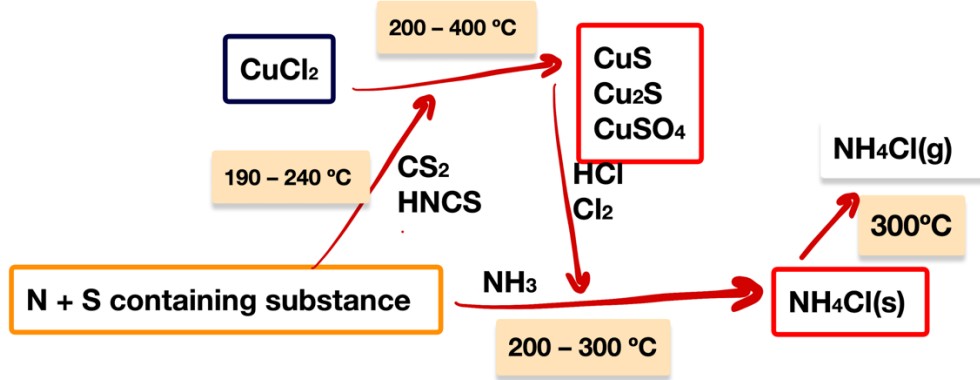

**Figure 3.** Pathways of inhibition of PCDD/Fs formation from N- and S- (adapted from [43]).

　　Regarding the difference between nitrogen and sulfur in the results of inhibition of the formation of PCDD/Fs, some authors have observed that nitrogen compounds are somewhat more effective in inhibiting the formation of PCDDs, while sulfur compounds act to a greater extent on the inhibition of PCDFs [15,28,34]. Hence, compounds containing nitrogen and sulfur at the same time are preferred. Ogawa et al. [34] studied the effect of adding $SO_2$ and mixing coal with an important sulfur content in the combustion of wood sawdust with PVC in a small fluidized bed incinerator, and observed a significant reduction in PCDD/Fs, being somewhat more important for PCDFs. Hajizadeh et al. [28] studied the effect of adding $SO_2$ and $NH_3$ on a laboratory scale model ash, at 225 and 375 °C (simulating the post-combustion zone), and observed, as did Zhan et al. [40], that temperature plays a very important role in the behavior of inhibitors (in addition to the concentration used and the point where they are added). $NH_3$ was more effective at the lower temperature, while the behavior of $SO_2$ was the opposite (reducing a higher proportion of PCDDs than $NH_3$ at the higher temperature). In the resulting solid phase (resulting fly ash), both had a greater effect than in the gas phase.

　　Mi Yan et al. [32] investigated the formation of PCDD/Fs produced from fly ash and its inhibition by ammonium sulfate and urea at different temperatures. The authors show that both inhibitors had a higher inhibition efficiency at high temperature (650 °C) than at low temperature (350 °C). Thus, the production of PCDD/Fs at 650 °C was quite small; a greater reduction in furan emissions (PDCFs) was observed, the most significant reductions being that of Tetra-CDF at low temperature (350 °C) and Octa-CDF at 650 °C. In a large-scale incinerator, chemical additives injected into the high-temperature section could enter the low-temperature area and continue to inhibit dioxin formation; therefore, more work is needed to evaluate different inhibitors for PCDD/Fs formation and determine the optimum injection temperature.

　　Marie-Rose et al. [44] suggested that the presence of $NH_3$ favors the formation of coke while decreasing the formation of $CO_2$. At a relatively low temperature (approx. 300 °C), $NH_3$ has been found to adsorb at strong acid sites, thereby reducing the oxidation of organic molecules to carbon dioxide. Furthermore, the formation of PCDD/Fs could also be reduced due to the blocking of these strong acid sites.

　　In their work, Fu et al. [18] studied three distinct –S- and $–NH_2$- or $NH^+_4$-containing compounds, including ammonium thiosulfate (ATS), amidosulfonic acid (ASA), and thiourea (TUA). The results revealed that the inhibition competency of the combined mixtures of S and N containing substances was strongly influenced by both the nature of the functional group of nitrogen and the value of the molar ratio (S + N)/Cl. In this way, the inhibition efficiency classifies as follows: TUA (99:8%) > ASA (92:4%) > ATS (85:4%), with the same sequence as PCDD/Fs.

　　On the other hand, Ma et al. [31] studied the effect of the presence of calcium oxide (CaO) and elemental sulfur (S) on the formation of PCDD/Fs during the municipal solid waste combustion process in laboratory runs performed at 800 °C. They concluded that

both substances could primarily consume chlorine sources or weaken chlorination in the PCDD/F formation process to slow down PCDF formation, but the inhibition mechanisms of sulphur and calcium oxide were different. Comparing the inhibitory effect of S with that of CaO, the inhibitory effect of S on the formation of hepta-congeners (HpCDD/Fs) was more remarkable, around 88.1%; on the other hand, CaO is especially active in the inhibition of the formation of hexa-congeners of dioxins and furans more clearly than sulfur, and the inhibition was 85.1%.

Wang et al. [36] studied the inhibitory effects of four amino compounds on the formation of chlorobenzenes (CBzs) in de-novo experiments, finding that the inhibitory effects follow the order $NH_4H_2PO_4 > NH_4HF_2 > (NH_4)_2SO_4 > NH_4Br$ under air flow, and are slightly different under nitrogen flow.

Soler et al. [35] also studied the mechanism implied in the thermal decomposition of model fly ash modified by the presence of N- and S-containing compounds. Specifically, the authors test the presence of thiourea (TUA), ammonium thiosulfate (TSA), and amidosulfonic acid (ASA) on the reactivity of fly ash air; however, the investigation was done using a thermobalance at different heating rates (5, 10, and 20 K min$^{-1}$) and analyzing the evolved gases. Their first conclusion is that the presence of the inhibitors, surprisingly, accelerated the decomposition of the model fly ash, i.e., the thermal decomposition took place at lower temperature (approx. 80 to 100 °C less) in the presence of TUA, TSA, or ASA. These authors assume that the increase in the decomposition rate is due to the existence of a reaction, not well defined, between the compounds containing N and S and the carbonaceous material.

## 3. Use of Other Sulphur and Nitrogen Containing Wastes to Reduce Emissions

In recent years, the possibility of using waste rich in nitrogen and/or sulfur as a source of inhibitor compounds has been considered, and thus the reduction of the emissions of pollutants while the waste is treated. Various authors, such as Amand et al. [22], showed the possibility of using waste mixtures to reduce pollutant emissions. The effects of the addition of urban water treatment sludge have been especially studied, since they generally have a high content of nitrogen compounds [45]. Chen et al. [24,25] used the drying gas of these sludges as a reaction atmosphere in the thermal decomposition of model samples, demonstrating that these gases are capable of suppressing the de novo synthesis of PCDD/Fs with an efficiency greater than 99% for some congeners, especially for dioxins versus furans.

Other nitrogenated wastes that have been used for inhibition are polyurethanes (mainly from mattresses). Polyurethanes are polymers made from long chain polyethers or polyesters covalently linked by a urethane bond (-NH-COO-). The manufacture of these plastics involves the reaction between polyfunctional isocyanates and alcohols [46]. Flexible polyurethane foams (FPUF) are mainly used in the manufacturing of upholstered furniture and mattresses.

Thermal degradation of different types of polyurethanes has been studied in the past. Boettner et al. [47] analyzed the volatiles produced during the combustion of four types of polyurethanes from car seats. They verified that the major products are CO and $CO_2$, hydrogen cyanide, acetaldehyde, and methanol, and they showed that, as the atmosphere becomes more oxidative, the amount of methanol and HCN produced decreases. In previous research, the pyrolysis and combustion of FPUF residues from mattresses was carried out, showing that the presence of oxygen effectively reduces the emission of PAHs and chlorobenzenes, as well as the $NH_3$ emission [48]. This decrease in ammonia has been attributed to the fact that the emission of PCDD/Fs simultaneously increases, due to the inhibitory effect shown by ammonia in the formation of these compounds [14,48].

Moreno et al. [33] tested the influence of the presence of another N-containing waste, as is polyurethane foam (PUF), in the formation of PCDD/Fs during the combustion of a PVC-containing waste. The authors also compared the presence of PUF in the waste itself with the effect of the presence of PUF pyrolysis gases in the decomposition of the PVC-containing waste. They found that the first alternative reduces the formation of

PCDD/Fs by 85.7% and dl-PCBs by 81.2% in WHO-TEQ toxicity, with the direct addition of PUF waste being the simplest one.

*Inhibition by Using Thermal Decomposition Products from Sewage Sludge*

The effect of adding sludge from urban sewage treatment plants in three variants has been especially studied: directly mixing the waste [22,37], using decomposition gas from previously dried sludge [25,49], and using the decomposition gas of the sludge together with other inhibitors such as thiourea [39].

Amand et al. [22] observed an important decrease in the emission of PCDD/Fs during co-combustion of solid recovered fuel (SRF) with municipal sewage sludge compared to when SRF was fired without SS as additional fuel. Also, the injection of ammonium sulphate was effective in this reduction, and the authors corroborate the presence of an 'alkali track' relating the level of alkali chlorides in the gas phase, the chlorine content in the deposits in the convection pass, and the PCDD/F formation.

Amand et al. [22] detected that direct mixing of waste can considerably reduce the levels of PCDD/Fs that are emitted. For this, they used an SS with a low dioxin content in a proportion of 8% together with 73% of wood and 19% of a waste of wood, textiles, paper, and plastic (which had a dioxin content higher than that of sludge). Experiments were carried out in a reactor on a scale close to the industrial one and under unique conditions, around 870 °C and with a slight excess of air with respect to the stoichiometric. Although the sludge contained transition metals that could favor the formation of dioxins, the dioxin emission was reduced by 86.6%. Xiao et al. [37] also observed a decrease in dioxin levels when mixing sewage sludge with sawdust during combustion; however, Zhang et al. [38] observed the opposite effect when mixing sludge with coal in a thermal power plant, which is why a more in-depth study is necessary. On the other hand, Chen et al. [49] verified that the sludge drying gas (actually slight thermal decomposition at about 300 °C of sludge previously dried at 105 °C in a different device) could inhibit the formation of dioxins in a fly ash model equivalent to those obtained in incineration plants of municipal waste, with an efficiency of up to 97.6% by weight, a result similar to that obtained with thiourea. The main compounds of the sludge drying gas were $NH_3$ and $SO_2$, although there may be other interesting nitrogen and sulfur compounds that also act as inhibitors and have not been determined. The whole experiment was carried out at 300 °C, using three types of sludge with nitrogen content around 4% and sulfur around 1%, except for one of the sludges that presented a value of 4%, although no important differences were observed with different contents in S. The doses used were 0.5–4 g of dry sludge for every 2 g of ash, and a flow of 300 mL/min of a gas was passed through them that contained 12% of $O_2$ in $N_2$. In a later work, Chen et al. [25] studied the effect of the dose of sludge, the temperature of the previous treatment of this sludge (250–350 °C), and the oxygen content (0–12%). They observed an efficiency greater than 99% of inhibition in terms of toxicity (higher for dioxins than for furans), with 300 °C being the most suitable temperature, and with a very low influence of the oxygen content. In different experiments, directly injecting $SO_2$ and $NH_3$, they established that $SO_2$ was more effective than $NH_3$ (61.9% and 38.6%, respectively). Zhan et al. [39], on the other hand, used the sludge decomposition gas obtained in a similar way to Chen et al. [24] in model fly ash, but in this case at a high temperature (650–850 °C), where the formation of PCDD/Fs could take place in the homogeneous gas phase by means of phenoxy radicals with chlorine atoms, instead of by synthesis de novo. In this case, the inhibition of dioxin formation was also observed with an efficiency of 90%. Zhan et al. [40] obtained similar inhibition results in urban solid waste incineration experiments in a pilot plant, where they added the decomposition gas of sewage sludge after combustion together with a certain amount of thiourea.

It should be noted that the experiments of Chen et al. [24] were made starting from sludge previously dried at 105 °C, which is how this residue is normally available in incineration plants, since the high moisture content of the sludge from the WWTP does not allow direct combustion.

Gomez–Rico et al. [50] studied the drying of SS at temperatures between 80 and 120 °C. In all cases, thermal drying resulted in the formation of volatile organic compounds (VOCs), including many PAHs, and prevent the need to control the emissions from the drying plants. This also indicates that the gases from SS drying can already contain a great amount of S and N gaseous compounds. In this context, Chen et al. [24] show the efficiency of SS drying gases for the inhibition of PCDD/Fs formation, in runs performed over model fly ashes, where they observed suppression of the 2,3,7,8-substituted PCDD/Fs formation. The authors calculate an efficiency up to 97.6% in wt. units and 96% in I-TEQ. Statistical treatment of the data showed that $NH_3$ and $SO_2$ are the most important components of the SS drying gases.

The sulfur compounds detected by Liu et al. [51] in pyrolysis are $H_2S$, $SO_2$, COS, $CS_2$, and $CH_3SH$ (from aliphatic and aromatic organic sulfur compounds present in the sludge). Other organic compounds in the sludge such as sulfonic acid and thiophene do not influence too much, and inorganic compounds such as sulfates and sulfites also remain stable at the mentioned temperatures. Regarding the nitrogen compounds also detected during sludge pyrolysis by Tian et al. [52], $NH_3$, HNCO, NO, and HCN were found, as well as compounds with amino and nitrile groups, and other heterocyclic nitrogen compounds (although these at somewhat higher temperatures). These compounds come mainly from the decomposition of proteins, amines, pyridines, and pyrroles.

Zhong et al. [41] studied the PCDD/F levels and phase distributions in a full-scale municipal solid waste incinerator with co-incinerating sewage sludge. The presence of a selective reduction catalyst (SCR) was also investigated. Their primary findings are that the presence of up to 5% of SS in the fuel did not increase the PCDD/Fs emission, and that the emission in all phases decreased significantly, including the condensed water collected during the sampling.

Chen et al. [25] also studied the inhibition of PCDD/Fs by SS decomposition gases. These authors observed suppression on the formation of toxic TCDD/Fs when the decomposition takes place in the presence of gases evolved from dried SS. The efficiency was close to 100%, being slightly higher for PCDDs than for PCDFs. Chen et al. also found that $SO_2$ was more effective (61.9% suppression efficiency) than $NH_3$ (38.6%) in suppressing PCDD/Fs formation.

On the other hand, Liu et al. [51] investigated the emission of organic sulfur in the gases during SS pyrolysis, showing that both aliphatic and aromatic sulfur compounds are present and that the transformation of aliphatic sulfurs into $H_2S$ begins at a relatively low temperature (250 °C). Also, aromatic sulfur transforms into $H_2S$, but the phenomenon was observed at higher temperature (350–450 °C).

Tian et al. [52] observed that the pyrolysis of SS is produced by the thermochemical decomposition of pyridine-N and pyrrole-N, and that fewer amine-N compounds are produced during pyrolysis. These authors also show that the heating rate of pyrolysis does not change the composition of the gases produced, but does produce significant changes in the species of liquid organic compounds produced.

Chen et al. [26] investigated the role of SS during the decomposition of fly ashes coming from municipal solid wastes. They found that the presence of SS inhibited the formation of PCDD/Fs by more than 96% and looked for the mechanisms of this inhibition. Treating the PCDD/Fs congener profiles of the gas emissions, they found that the SS suppressed formation through de-novo pathway, the suppression of the chlorophenol route being less important. Also, these authors determined that the main mechanism for the inhibition is the cleavage of the oxygen bridge in the molecules.

On the other hand, Zhang et al. [38] studied the emission of dioxins and furans in the combustion of coal in power plants, comparing the effect of different proportions of SS (5 to 20 wt.%). Surprisingly, the authors indicate that the amount of PCDD/Fs increased as the amount of SS increased in the fuel, when considering the whole mass balance. Nevertheless, authors demonstrate that co-combustion of SS is a sink for PCDD/Fs, as mentioned also by

Xiao et al. [37]. The main output current of these pollutants is fly ash, with a comparable amount to the output by emitted gas.

Xiao et al. [37] show that there is a predominance of furans (PCDFs) over dioxins (PCDDs) in all the output currents (stack gas, fly ash, slag) during the co-combustion of SS and wood sawdust.

Recently, a study was performed on the emissions of pollutants from the thermal decomposition of PVC and sewage sludges in different experimental conditions [27]. In this work, the materials were mixed in different proportions (ionic ratios "Ri" of (N + S) to chlorine equal to 0.25, 0.50 and 0.75) in addition to the combustion of the materials separately, and were reacted with oxygen at 850 °C. Oxygen was also present in different proportions, to study the effect of the presence of this element. Partial combustion reactions were performed with values of λ (excess oxygen) equal to 0.15 and 0.50. PAHs and PCDD/Fs were evaluated. Figure 4 shows the inhibitory effect of the presence of SS in the emission of PAHs, in all runs performed. This inhibitory effect is calculated comparing the emissions to that of the materials decomposed separately.

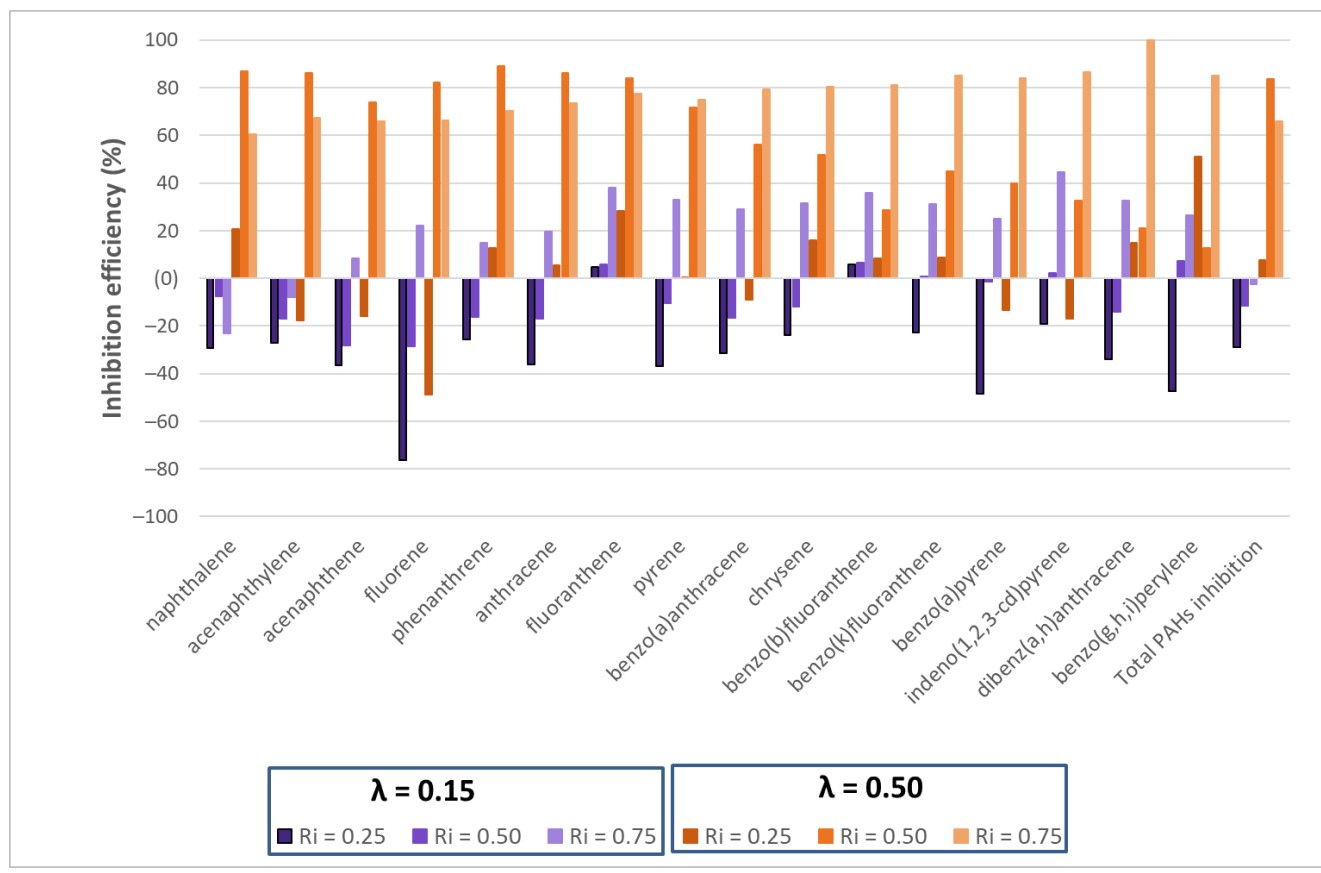

**Figure 4.** Inhibition efficiency (%) in PAH formation for every experiment run at 850 °C and both oxygen ratios: λ = 0.15 and 0.50.

Figure 4 shows that the production of PAHs is much higher at a low oxygen ratio (λ = 0.15), as expected, due to a less efficient combustion [53]. Concerning the presence of SS, the concentration of PAHs present in the sample decreased notably when the inhibition ratio increased, with some of the compounds being reduced by more than 80% (Figure 4). In that behavior, we found that λ = 0.50 was always better to reduce the emission of PAHs, therefore, in the presence of more oxygen, the formation of PAHs is lower, with Ri= 0.50 being the mixture that showed the best results.

For the PCDD/Fs emissions, the results obtained at different experimental conditions are outlined in Figure 5. The results obtained for PCDD/Fs at λ = 0.15 were different from those expected. Intermediate values were expected between emission of SS and PVC e-waste. For the hexa-, hepta-, and octachlorinated congeners, the mixture Ri = 0.25 presented higher dioxin emissions than expected due to the composition of the sludge, composed by around 6% of iron, among other metals in much less proportion, such as copper (0.09%). Previous studies showed that the presence of these metals catalyzes the formation reaction of PCDDs and PCDFs [12,13]. According to Fujimori et al. [54], some of the active catalysts could already be present in the sample.

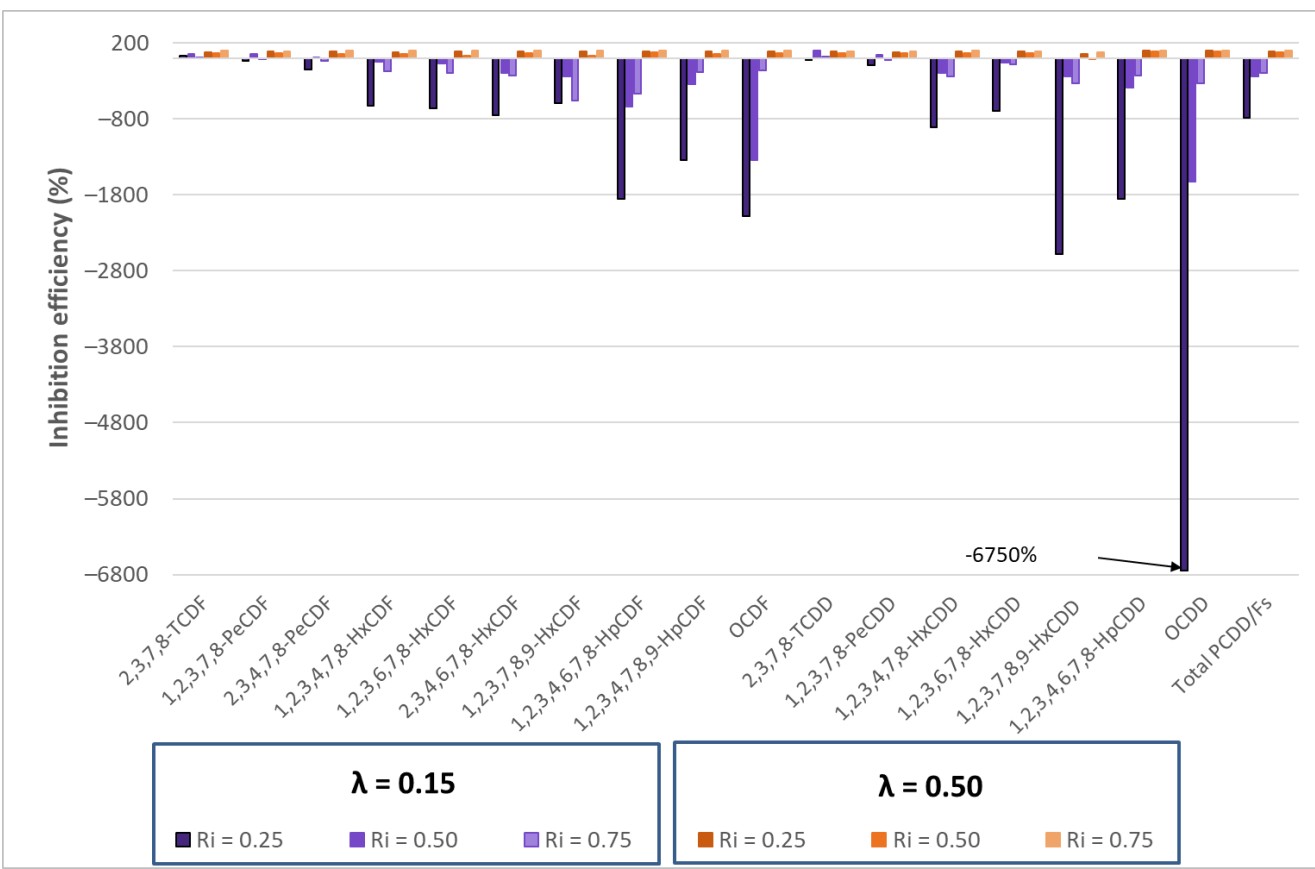

**Figure 5.** Inhibition efficiency (%) in PCDD/Fs formation for every experiment run at 850 °C and both oxygen ratios λ = 0.15 and 0.50.

For the low oxygen ratio, λ = 0.15, the lowest emissions were obtained for an Ri of 0.75 and the highest emissions for Ri = 0.25, i.e., a higher amount of SS produced higher inhibition. When oxygen is fed at λ = 0.50 (greater presence), the lowest emissions were obtained also for an inhibition ratio of 0.75.

It can be observed that, when working with small inhibition ratios (Ri = 0.25) or with high inhibition ratios (Ri = 0.75), λ = 0.50 was always better to reduce the concentration of PCDD/Fs. However, for intermediate inhibition ratios around 0.50, the emissions obtained were similar for both oxygen ratios.

As a conclusion, Ri = 0.75 was always better to minimize the emission of PCDD/Fs, and, therefore, in the presence of more SS, the formation of PCDD/Fs is much lower for both oxygen atmospheres, with λ = 0.50 being the oxygen ratio that clearly showed the best results in terms of the inhibition efficiency.

## 4. Conclusions

This review presents the main research that is being carried out regarding the use of nitrogenous and sulfurized substances in the reduction of pollutant emissions in incineration furnaces.

Different chemicals have been tested in the literature to reduce the emission of different POPs, especially those with a halogenated structure, due to their dangerousness. In this sense, substances such as thiourea, ammonium sulfate, or sulfamidic acid prevent the formation of PCDD/Fs with an efficiency higher than 90%.

Also, other substances containing nitrogen and sulfur, including wastes, appear to be a promising way to reduce persistent organic compound emissions by very high percentages. In this sense, sewage sludges seem to be a very interesting waste, as they are very abundant in urban centers and present both nitrogen and sulphur in their chemistry. In addition, the combustion of municipal waste together with sewage sludge avoids an important problem, the disposal of these sludges, which in many cases has high metal content, making them unsuitable for agricultural amendment.

The data presented in this review indicate that the joint combustion of sewage sludge and municipal waste reduces the emissions of dioxins, furans, and related species by more than 90%. This should translate into legislation requiring municipalities to add SS at MSWI plants.

Regarding the direction of future research, it is necessary to verify the efficiency of the elimination of other polluting species, particularly those that present bromine in their structure and that are generally related to the combustion of electrical and electronic materials (since bromine is used as a flame retardant in this equipment).

**Funding:** This research was funded by MINISTRY OF SCIENCE AND INNOVATION (SPAIN), grant number PID2019-105359RB-I00.

**Institutional Review Board Statement:** Not applicable.

**Informed Consent Statement:** Not applicable.

**Conflicts of Interest:** The author declares no conflict of interest.

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
