# Peer review of "Sewage Sludge as Inhibitor of the Formation of Persistent Organic Pollutants during Incineration"

_sustainability, doi:10.3390/su131910935_

Round 1

Reviewer 1 Report

I decided it was one of important reviews.

Author Response

Thank you for your comments.

Reviewer 2 Report

Sewage sludge as inhibitor of the formation of persistent or-2 ganic pollutants during incineration

  1. Please revise the abstract, introduction part is very long. Give more space to discuss data collection, finding and conclusions. Also highlight the contribution part
  2. Avoid using abbreviation in keywords.
  3. Figure 1 the font is very small, unreadable. Any reference?
  4. Ata the end of introduction part, please highlight the gap of information and the main contribution of this article.
  5. Conclusion section should be strengthened and extended. Currently very limited.
  6. Based on this review what is the recommendation and future research direction

Author Response

COMMENTS & REPLIES ## REVIEWER 2 ##

  1. Please revise the abstract, introduction part is very long. Give more space to discuss data collection, finding and conclusions. Also highlight the contribution part.

Reply:

Thanks for this suggestion. Abstract has been rewritten, eliminating the introduction and pointing out specifically some conclusions. In the new version of the Abstract, the following can be read: “With the objective to suppress dioxins and furans (PCDD/Fs) emission in municipal solid waste incineration plants (MSWI), different chemical inhibitors have been tested. Among these inhibitors, nitrogen and sulphur compounds can significantly suppress PCDD/Fs formation via de novo synthesis, which very interesting results with very little capital investment. In recent years, the possibility of using waste rich in nitrogen and/or sulphur as a source of inhibitor compounds has been considered, and thus reduce the emissions of pollutants while the waste is treated. The effect of adding sludge from urban sewage treatment plants in three variants has been specially studied: directly mixing the waste, using the decomposition gas of the previously dried sludge, and using the decomposition gas of the sludge together with other inhibitors such as thiourea. Reduction of emission in laboratory tests using model samples indicated the efficiency to be higher than 99 %, using sewage sludge (SS) as inhibitor, whereas in actual MSWI plants the efficiency can be as high as 90 %.”

  1. Avoid using abbreviation in keywords.

Reply:

Thanks for this suggestion. The abbreviations that have been introduced in the keywords are widely used in all publications related to this subject and we think that they can help to find the work in the corresponding databases. However, if the publisher deems it necessary, they could be replaced by their corresponding full names.

  1. Figure 1 the font is very small, unreadable. Any reference?

Reply:

Thanks for this suggestion. Figure 1 has been redrawn and a reference has been introduced.

  1. At the end of introduction part, please highlight the gap of information and the main contribution of this article.

Reply:

Thanks for this suggestion. The introduction, in the new version of the manuscript, ends: “In the present work, a review of data found in literature about inhibition with chemicals in MSWI plants is done, with particular interest in the use of sewage sludge as as a material that provides sulfur and nitrogen, and investigating the mechanism of action of these compounds.”

  1. Conclusion section should be strengthened and extended. Currently very limited.
  2. Based on this review what is the recommendation and future research direction

Reply:

Thanks for these suggestions. Conclusions have been re-written with special focus on the data presented. Also, recommendation and future research has been commented. (please, consult the new version of the manuscript).